# MCount: An automated colony counting tool for high-throughput microbiology

**Sijie Chen**[1,2], **Po-Hsun Huang**[1], **Hyungseok Kim**[1,2], **Yuhe Cui**[1], **Cullen R. Buie**[1]*

**1** Department of Mechanical Engineering, Massachusetts Institute of Technology, Cambridge, Massachusetts, United States of America, **2** Institute for Data, Systems, and Society, Massachusetts Institute of Technology, Cambridge, Massachusetts, United States of America

* crb@mit.edu

## Abstract

Accurate colony counting is crucial for assessing microbial growth in high-throughput workflows. However, existing automated counting solutions struggle with the issue of merged colonies, a common occurrence in high-throughput plating. To overcome this limitation, we propose MCount, the only known solution that incorporates both contour information and regional algorithms for colony counting. By optimizing the pairing of contours with regional candidate circles, MCount can accurately infer the number of merged colonies. We evaluate MCount on a precisely labeled *Escherichia coli* dataset of 960 images (15,847 segments) and achieve an average error rate of 3.99%, significantly outperforming existing published solutions such as NICE (16.54%), AutoCellSeg (33.54%), and OpenCFU (50.31%). MCount is user-friendly as it only requires two hyperparameters. To further facilitate deployment in scenarios with limited labeled data, we propose statistical methods for selecting the hyperparameters using few labeled or even unlabeled data points, all of which guarantee consistently low error rates. MCount presents a promising solution for accurate and efficient colony counting in application workflows requiring high throughput, particularly in cases with merged colonies.

## 1. Introduction

Quantitative assessment of microorganisms is a critical procedure in the field of microbiology, and various methods have been developed to estimate microorganism population levels, including quantitative PCR [1], flow cytometry [2], spectrophotometry [3,4], and colony-forming units (CFU) counting [5,6]. Among these methods, CFU counting is the oldest and most widely used method, and its efficacy and reliability have been examined since the 1880s [6,7]. In addition, its simple operational protocol and minimal consumables cost have solidified CFU counting as the gold standard in microbiology.

However, traditional manual CFU counting is a time-consuming and labor-intensive process when dealing with many images (e.g., more than one hundred). Thus, researchers have devoted considerable effort to developing numerous solutions over the decades to realize an easier, faster, more accurate, and reliable counting method. Currently, popular solutions include commercial products such as SphereFlash (IUL Instruments) and ProtoCOL 3 (Synbiosis), as well as open-source tools based on various counting algorithms [8–18]. While

**Data availability statement:** The source code and integrated solution are freely available at

https://github.com/hyu-kim/mcount. All methods implemented and data used are publicly available on Dryad at https://doi.org/10.5061/dryad.2280gb62f

**Funding:** This work was supported by National Institutes of Health (NIH) through grant number RM1 GM135102 to C.R.B., S.C., P.-H.H., and H.K. received support from the Department of Energy's Genome Sciences Program through grant SCW1039. URL1: https://grantome.com/grant/NIH/RM1-GM135102-01 URL2: https://genomicscience.energy.gov/llnl/ The funders had no role in study design, data collection and analysis, decision to publish, or preparation of the manuscript.

**Competing interests:** I have read the journal's policy and the authors of this manuscript have the following competing interests: C.R.B. is a Co-Founder and Advisor of Kytopen Corp. The remaining authors declare no competing interests. This does not alter our adherence to PLOS ONE policies on sharing data and materials. There are no patents, products in development or marketed products associated with this research to declare.

commercial products are usually easy to operate, they are expensive and highly specialized for specific counting scenarios, and their proprietary programming nature makes them difficult to modify or share. In contrast, open-source tools are gaining more attention, especially those capable of batch processing [9–11]. NIST's Integrated Colony Enumerator (NICE) is a tool that has been popular since 2009, which is based on the combination of extended minima function and thresholding algorithms [9]. It has a relatively short image processing time (< 5 seconds per image) and a friendly user interface. OpenCFU, published in 2013, is a tool claiming to be faster, more accurate, and more robust than NICE [10]. It adopts the watershed algorithm along with a series of pre- and post-processing filters. AutoCellSeg was developed in 2017 [11] to reduce the hyperparameter selection effort during counting, based on a feedback-based watershed algorithm, and it has an interactive graphical user interface that is appealing to those not familiar with programming.

The existing colony-counting solutions work well on single petri dish plates of $60\,cm^2$, where 25 ~ 250 colonies are randomly distributed, making colony merging issues rare [7,19,20]. Most colony-plating protocols require careful selection of the serial dilution ratio to minimize the chance of merged colony. However, modern colony-counting requires higher throughput, and a typical example is plating samples from a 96-well plate on a single-well rectangular plate of $109\,cm^2$ (Fig 1a). In this case, more than 10 colonies are plated on an area less than $1\,cm^2$ for each sample, resulting in >10 times higher density, which often leads to merged colonies. Furthermore, in such high-throughput workflows, the smaller area results in fewer image pixels per sample, reducing image quality. These difficulties lead to significant counting underestimation with current solutions (Fig 1b). Although NICE can recognize most single colonies, it tends to count merged colonies as one. AutoCellSeg typically counts

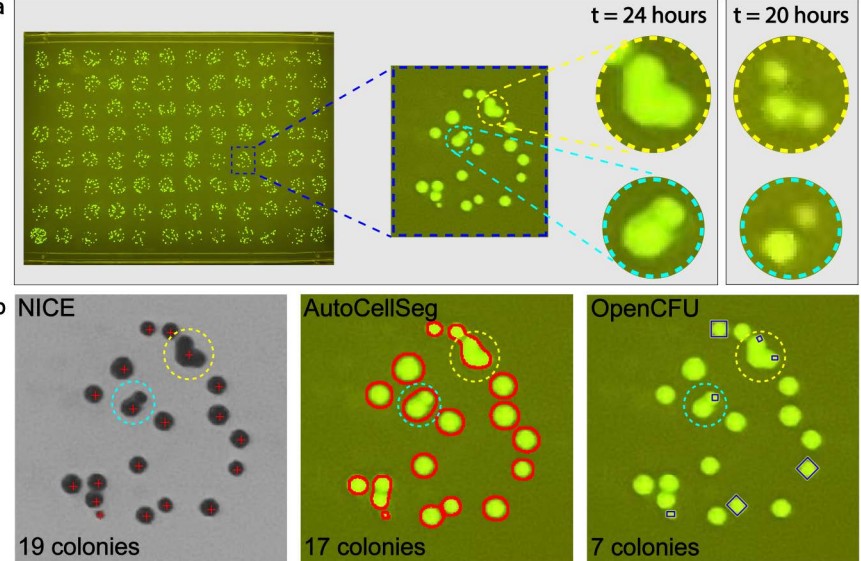

**Fig 1. The need for a more powerful colony counting algorithm. (a)** Colonies of fluorescent *E. coli* NEB10-beta wereplated on agar in an $8 \times 12$ array format. As the area of each well is small, colony merging occurs frequently and the sub-image in the blue rectangle is an example. While it is challenging to determine the exact number of merged colonies in the yellow and cyan circles, a skilled person can count 3 and 2 colonies, respectively, which can be verified by the photograph taken a few hours earlier. **(b)** NICE correctly counts all single colonies but counts also merged colonies as one [9]; AutoCellSeg uses the watershed algorithm to count every connected region as one colony, regardless of the region's shape [11]; OpenCFU applies sophisticated rules to count only perfect circle-shaped colonies but fails to recognize most single colonies [10]. All three algorithms underestimate the total number of colonies, which is 22.

each continuous region as one, regardless of the region's shape, so it cannot handle scenarios involving colony merging. As for OpenCFU, it fails to recognize most colonies, including single colonies, due to the lower image quality for each sample of the rectangular plate. The failure of these solutions results from the fact that they rely on region-based algorithms and do not take contour information into account.

Colony counting is a sub-domain of object counting that includes measurement of the number of biological cells [21–23], pollen grains in air [24,25], and bubbles for analyzing two-phase systems [26,27]. Classical object counting approaches can be categorized into contour-based or region-based methods. Contour-based methods aim to recognize the shape only from the contour pixels and use methods such as the Hough-transform [28,29] and least squares fit for circular contour [26,28,30]. They also involve algorithms such as concave point detection methods to split the contour into segments [27,30–33]. Contour-based methods can provide precise results given a high-definition image, but they typically require more computational resources. Region-based methods take all shape pixels into account, and classical methods include the extended minima function method [9,16,34], morphological operations [35], distance transformation [16,31,36], and the watershed algorithm [10,11,16,17,24,34]. Region-based methods can tolerate more noise and have faster processing speed, but their recognition accuracy is usually lower. To combine the advantages of both methods, some have made efforts to pair the extended minima method with contour segments [26,27,31,32]. In recent studies, optimization algorithms have been adopted [31,33], eliminating the computationally expensive exhausted pairing [26,27]. Surprisingly, although classical object counting algorithms are advancing rapidly in other sub-domains, colony counting lags behind with a reliance on region-based algorithms, leading to poor accuracy in high-throughput workflows.

In addition to classical object counting methods, neural networks have been applied across many fields due to their success in processing complex tasks [22,37,38]. However, when applied to colony counting, these methods encounter significant challenges, primarily due to the lack of standardized, large-scale, high-quality datasets. Most existing datasets are small, often only a few hundred images, which are insufficient for training deep learning models with numerous parameters. Additionally, neural networks often function as "black boxes", which may provide final counts without visually delineating individual colonies, a drawback for biologists who need visual confirmation. While advanced models could include features like contouring, they require more robust datasets and computational resources. For these reasons, we believe classical algorithms remain a more suitable choice for colony counting.

To ensure user-friendliness and operational simplicity, counting tools typically emphasize a few intuitive hyperparameters. However, while these tools focus more on the physical meaning of different hyperparameters, they often lack discussion on properly tuning these hyperparameters, i.e., the hyperparameter optimization problem, which is an essential topic in the field of machine learning to ensure high quality performance with minimal human effort [39,40]. Consequently, deploying such solutions to various counting tasks, even with minor differences, can be challenging. Therefore, there is a need for not only a better colony counting algorithm but also a consistent method for hyperparameter optimization.

In this paper we propose a new solution, MCount (Merged-colony Counting), that can precisely infer the number of merged colonies from relatively small images, meeting the demands of high-throughput colony counting. MCount employs both region-based and contour-based algorithms, leading to much higher accuracy than existing tools. Since there is no standard benchmark for high-throughput colony counting tasks, we constructed a dataset for performance evaluation. In addition, we address the hyperparameter optimization problem. Given that the number of labeled data is often insufficient, we propose novel statistical methods for hyperparameter optimization, ensuring a low error rate even with minimal labeled data or

even unlabeled data. Finally, we examine the statistical robustness of the proposed hyperparameter-tuning methods.

## 2. Methods

Fig 2 presents a detailed flowchart of MCount, depicting the process of foreground extraction, contour extraction, regional circle fitting, and optimization.

### 2.1. Algorithms adopted by MCount

**2.1.1. Foreground extraction.** The first step is to obtain connected (overlapping) colony segments by using a series of filters. The original image is binarized using Otsu thresholding, which separates image pixels into two categories, i.e., foreground and background, by maximizing inter-class variance [41]. Next, the connected component labeling algorithm is used to separate disconnected segments. Finally, morphological operations (erosion operation followed by a dilation operation) are applied to each colony segment twice to remove white noise in the image.

**2.1.2. Contour extraction.** The next step is to extract contours from each colony segment using a classic border following algorithm [42]. As each contour may correspond to the edges of several colonies, we need to split each contour into several segments, so that each piece only corresponds to one colony. We assume that any overlap between multiple colonies results in a

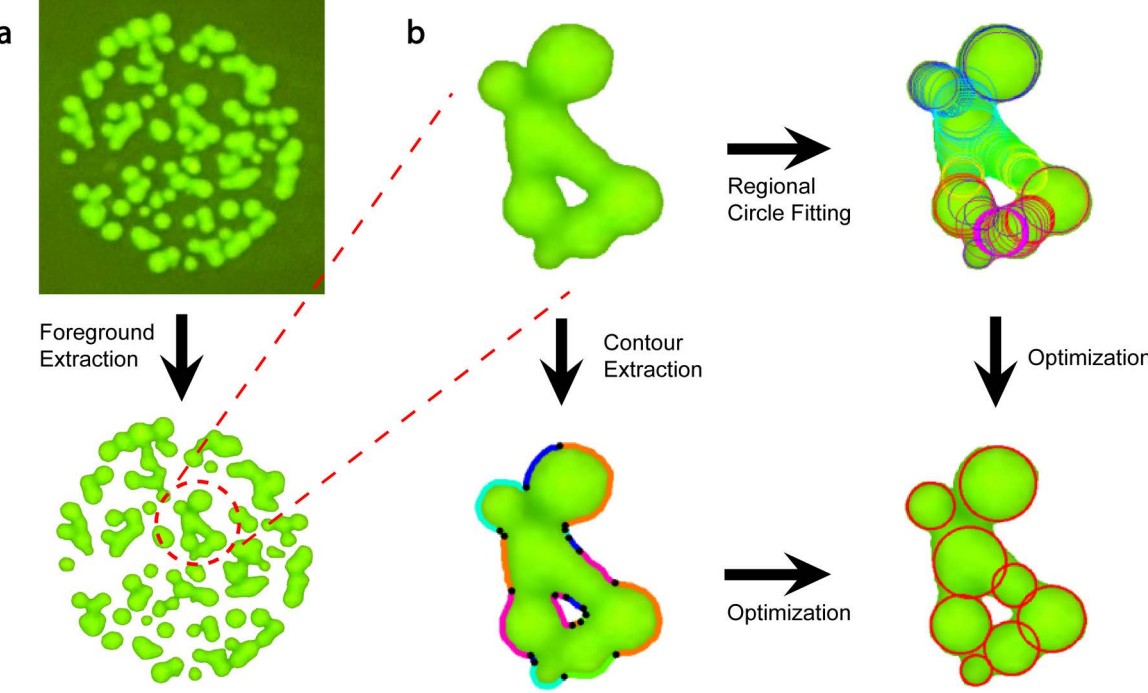

**Fig 2. MCount uses a combination of contour-based and region-based information to accurately count merged colonies. (a)** The foreground and background are separated using Otsu thresholding [41], and colonies are segmented into disconnected segments, which are then smoothed to remove noise. **(b)** Contour-based algorithms are applied to each colony segment to identify turning points (colored in black), which divide the inner and outer contours into contour segments represented by different colors. Meanwhile, region-based algorithms generate candidate circles denoted in different colors that can fit the boundary well. Finally, proper circles are selected from candidate circles based on the shape of all contour segments. This selection process is converted into an optimization problem that is solved to correctly recognize merged colonies.

shape with concave edge points that correspond to the intersections of the colony boundaries, which allows us to use the concave points to split the contour. To identify the concave points, we use an algorithm based on the polygon approximation algorithm [27] and the algorithm proposed by Zafari *et al.* [43], and the pseudo-code is provided in Algorithm 1.

In brief, as shown in lines 1-18, the contour is approximated using polynomial fitting, where all the vertices of the polygon are on the contour. Such vertices are called turning points, denoted as $T$, and hyperparameter $d$ controls how densely the turning points are selected from the contour, as shown in S1 Fig. For example, when $d$ increases, fewer turning points are selected. Next, all concave vertices of the polygon are further selected from the turning points by implementing code from lines 19-24. These concave points divide the contour into several segments, and each piece is denoted by a different color in Fig 2b.

**Algorithm 1. Contour segmentation based on concave point detection.** Given a contour $C$ with a hyperparameter $d$ that determines contour fineness, this algorithm first approximates $C$ using polynomial fitting and selects turning points $T$ from the polygon approximation. Then, it identifies concave points $Conc$ from the turning points $T$ to split the contour into several segments.

---

**Algorithm 1:** Extracting concave points of a contour

**Input:**
$C$: list of contour pixel points; $n$ points in total
$d$: adjustable parameter that defines the fineness of contour

**Output:**
$T$: list of turning pixel points; $m$ points in total
$Conc$: list of concave pixel points

1 $T.append(C_0)$
2 $i \leftarrow 0, \quad k \leftarrow 2$
3 **while** $k < n$ **do**
4 $d_{max} \leftarrow -inf$
5 **for** $j \leftarrow i + 1$ **to** $k$ **do**
6 $dist \leftarrow$ the distance from point $C_j$ to line $\overline{C_i C_k}$
7 **if** $dist > d_{max}$ **then**
8 $d_{max} \leftarrow dist$
9 $ind \leftarrow j$
10 **end**
11 **end**
12 **if** $dist > d$ **then**
13 $T.append(C_{ind})$
14 $i \leftarrow ind, \quad k \leftarrow ind + 2$
15 **else**
16 $k \leftarrow k + 1$
17 **end**
18 **end**
19 $Conc.append(T_0)$
20 **for** $i \leftarrow 1$ **to** $m - 1$ **do**
21 **if** $Angle(\overrightarrow{C_{i-1}C_i}, \overrightarrow{C_i C_{i+1}}) < 0$ **then**
22 $Conc.append(T_i)$
23 **end**
24 **end**

---

**2.1.3. Regional circle fitting.** In this step, a set of candidate circles is generated from each colony segment. Two types of circles are considered. The first type of circle is identified using a region-based algorithm, where distance transformation is applied to the binary foreground image so that the center of circles is the local maxima of distance [44]. The other type of circle

is obtained by a least square fit of each split contour piece [27]. In Fig 2b, all the candidate circles are denoted by different colors.

**2.1.4. Optimization.** The final step of the algorithm is to pair the split contour segments with candidate circles, so that each contour will only match one circle. Denoting $C_i$ as the $i$ th contour piece and $O_j$ as the $j$ th circle (the edge of the circle rather than the center), the pairing problem that we solve is:

$$X_{ij} \begin{cases} 1 & if \ allocating \ C_i to \ O_j \\ 0 & otherwise \end{cases} \tag{1}$$

$$Y_j = \begin{cases} 1 & if \ any \ contour \ is \ allocated \ to \ O_j \\ 0 & otherwise \end{cases} \tag{2}$$

$O_j$ is a recognized as a colony only when $Y_j = 1$. To formulate this pairing problem, we define the distance $D_{ij}$ of allocating $O_j$ to $C_i$ as:

$$D_{ij} = \sum_p \min_q \overline{pq}, \ p \in C_i, q \in O_j \tag{3}$$

where $p$ and $q$ are the pixels of contour piece and circle, respectively, and $\overline{pq}$ represents the distance between pixel $p$ and $q$.

The smaller $D_{ij}$, the better the pairing of circle $O_j$ with contour piece $C_j$. Thus, an intuitive way to pair all contour segments with proper circles is to match every contour piece to the closest circle, i.e., we are looking for a pairing that could lead to a minimized total distance, which could be converted into an optimization problem. However, simply minimizing the total distance without any constraints will lead to overfitting because different contour segments from the same colony might be fitted to multiple adjacent circles. Therefore, some constraints must be considered, and an example is to control the total number of recognized circles, $\sum Y_j$, which can be converted to a zero-one integer programming problem with objective function $L(X_{ij}, Y_i)$:

$$(X_{ij}, Y_j) = \underset{X_{ij}, Y_i}{\mathrm{argmin}} \ L(X_{ij}, Y_i) = \underset{X_{ij}, Y_i}{\mathrm{argmin}} \sum_{i,j} D_{ij} X_{ij} + \lambda \sum_j Y_j \tag{4}$$

$$\mathrm{s.t.} \ \sum_j X_{ij} = 1$$

$$0 \le X_{ij} \le 1$$

$$Y_j \le \sum_i X_{ij} \le Y_j \bullet M$$

$$0 \le Y_j \le 1$$

where $\lambda$ denotes the strength of the constraint and $M$ denotes the total number of contour segments.

By solving equation (4), we can obtain all the recognized colonies $O_j$ with $Y_j = 1$.

**2.1.5. Reducing computational complexity of the optimization step.** We recognize that the most time-consuming part during the implementation of MCount is the optimization step

because the zero-one integer programming problem is an NP-complete problem whose worst-case runtime grows exponentially. To mitigate this issue, the segment separation procedure is introduced into the foreground extraction step, which can dramatically reduce the computational complexity of the optimization step by reducing the total number of contour segments to a few hundred.

In addition, as there are several approximation algorithms available [45,46], which are already integrated into widely adopted linear programming modelers, e.g., PuLP in Python, we choose one (PuLP solver "COIN_CMD") that yields a high recognition accuracy with relatively short computation time. The actual runtime of MCount is discussed in the Results and Discussion section.

## 2.2. Benchmark for performance evaluation

We first sought to evaluate the performance of MCount by using a benchmark dataset. As there is no standard benchmark available, existing colony counting tasks have relied on their own datasets [8–13]. However, we find that the datasets exhibited at least one of the following concerns when applied to MCount: (a) insufficient number of labeled images (typically less than 100); (b) too many colonies in each labeled image (more than 50), so that the counting number may no longer be the best criteria to evaluate recognition performance; (c) incorrect ground-truth labels, especially on highly merged colonies.

To address these issues, we have created our own benchmark dataset colonies of GFP fluorescent *E. coli* DH10-beta, a model organism designed for creating mutant libraries suitable for high-throughput purposes (S2 Fig). While our method is versatile and can effectively adapt to non-fluorescent imaging scenarios, the use of fluorescence imaging primarily facilitated the manual annotation process by providing clear and distinct colony boundaries, enabling more accurate ground-truth labeling essential for training and validating the model. The dataset includes 960 labeled images that are large enough to address the problem of an insufficiently large data set. These images were derived from 10 sets of 96-well plates, each with slightly different experimental conditions performed on different dates, ensuring that the dataset contains a certain degree of variation. These variations include differences in colony size, density, and the setup used during image collection, such as camera positioning and background lighting.

Additionally, we increased the size of our training set by decomposing these images into 15,847 colony segments using foreground extraction. Most colony segments have less than 5 colonies, which helps to address problem (b). To mitigate problem (c), we carefully labeled historical images in which the merged colonies were separated. The cell preparation procedure and more detailed discussions on problems (b) and (c) are included in the S1 File.

## 3. Results and Discussion

### 3.1. MCount only has two adjustable hyperparameters

Providing more hyperparameters can increase the flexibility of algorithm tuning, which often improves recognition accuracy. However, to prevent overfitting as well as to ensure user-friendliness, we aim to offer minimal adjustable hyperparameters by choosing the most important ones. In the case of MCount, we have identified two hyperparameters that significantly influence its accuracy: $d$ and $\lambda$, which control the contour extraction and optimization steps, respectively.

Hyperparameter $d$ represents the fineness of the polygon approximation fitting of the contour. When $d$ is small, the contour will have more turning points, resulting in more contour segments being generated, as shown in Fig 3a. However, setting too small $d$ should be avoided,

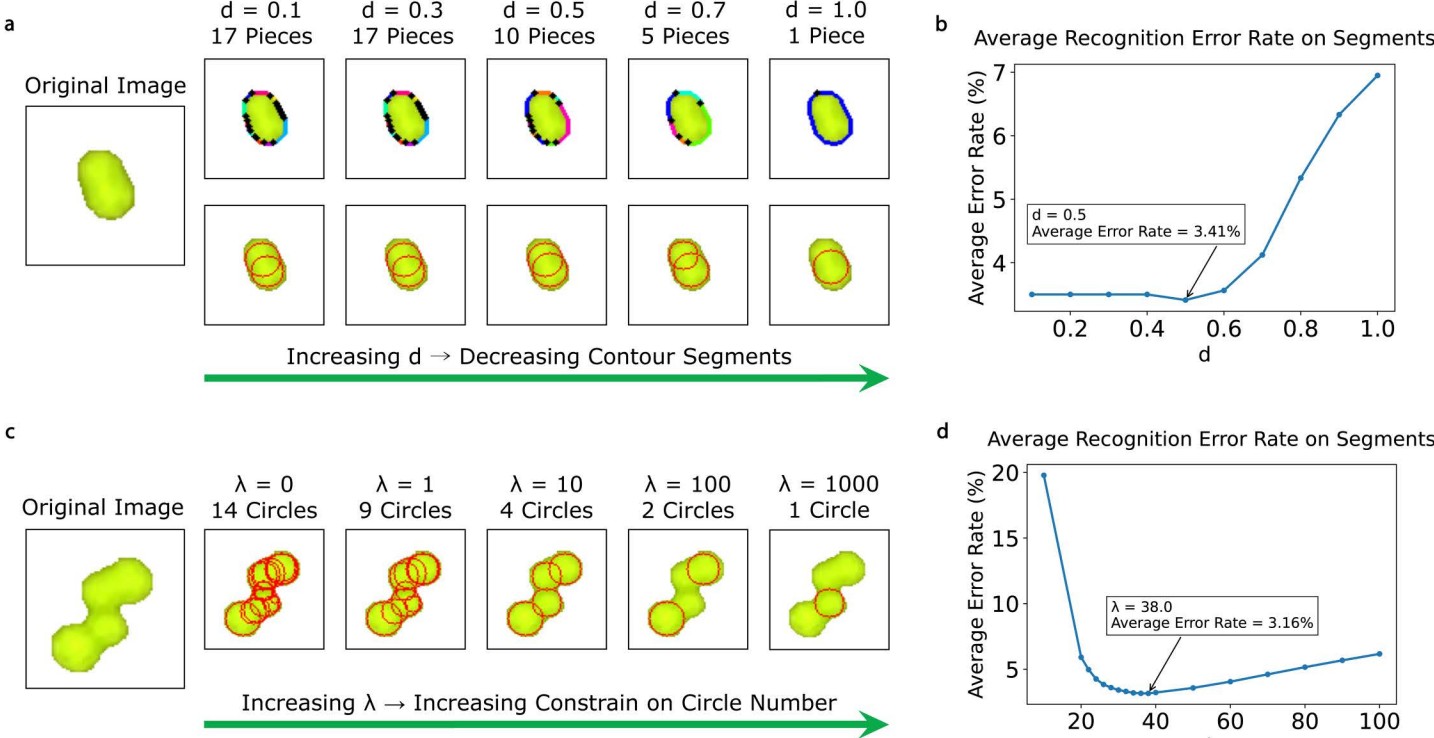

**Fig 3. MCount has two adjustable hyperparameters, *d* and *λ*, which control the contour fineness and constrain the circle number, respectively. (a)** The number of contour segments decreases as *d* increases, but excessively large *d* may lead to a failure to recognize merged colonies. The original image has two colonies. **(b)** The average error rate on the segment dataset versus *d* at $\lambda = 26$, where the minimum average error rate of 3.41% is achieved when $d = 0.5$. **(c)** A larger *λ* imposes a stronger constraint on the circle number. By tuning the *λ* value within the proper range, e.g., $\lambda = 10$, MCount can correctly recognize merged colonies. The original image has four colonies. **(d)** The average error rate on the segment dataset versus *λ* when $d = 0.5$, where the minimum average error rate of 3.16% is achieved at $\lambda = 38$.

because it will result in an unnecessarily large number of contour segments and candidate circles (each contour piece will generate one least-squares circle), leading to a longer runtime in the optimization step. This also reduces the algorithm's robustness because each contour piece will be too short to accurately calculate the curvature. Conversely, if *d* is too large, very few turning points will be used to fit the contour, leading to an underestimation of the colony count. To minimize the average recognition error on segments, we implemented different values of *d* on 15,847 colony segments, as shown in Fig 3b, and found that the best value for *d* is 0.5 (when $\lambda = 26$). The recognition error on a segment is defined as the difference between the number of colonies counted and the labeled number of colonies divided by the labeled number. Another way to define recognition error is to set 100% for a segment whenever the counted number differs from the label number. The plot for such a recognition error on segments is shown in S3 Fig (a).

The other important hyperparameter is *λ*, which is defined in Equation (4) and reflects the trade-off between colony number constraints and pairing of contour segments to candidate circles. As shown in Fig 3c, when $\lambda = 0$, there is no constraint on colony number, and almost every contour piece matches a unique candidate circle, even if some segments come from the same colony. When *λ* increases, the counted number monotonically increases, and $\lambda = 10$ provides a reasonable recognition result. When *λ* is very large, such as $\lambda = 1000$, the constraint is so strong that only one circle is drawn. We determined that the best value for *λ* to minimize the average recognition error on segments is 38 (when $d = 0.5$) by testing colony

segments. S3 Fig (b) shows the optimal values for $\lambda$ using the alternative definition for recognition error.

Both hyperparameters $d$ and $\lambda$ are critical to achieve accurate colony-counting results with MCount. However, their optimum values are dependent upon various factors such as image resolution and colony size, thus there is no universal set of values that will work for all scenarios. The standard approach for determining the optimal hyperparameter values is to optimize metrics such as the average recognition error through grid search and k-fold cross-validation, as demonstrated in the following section.

### 3.2. MCount performance

We utilized grid search and 10-fold cross-validation to minimize the average error rate on 15,847 colony segments and 960 sub-images to systematically determine the appropriate values for the hyperparameters $(d, \lambda)$. The optimal hyperparameter values were found to be $(d, \lambda) = (0.5, 38)$ and $(d, \lambda) = (0.5, 26)$, respectively. In this section, where we focus on the performance on sub-images, we will assess the performance of MCount using $(d, \lambda) = (0.5, 26)$.

Fig 4a shows a bubble plot of MCount's result versus the ground-truth label on colony segments. The regression line of MCount is close to the ground-truth line, indicating that

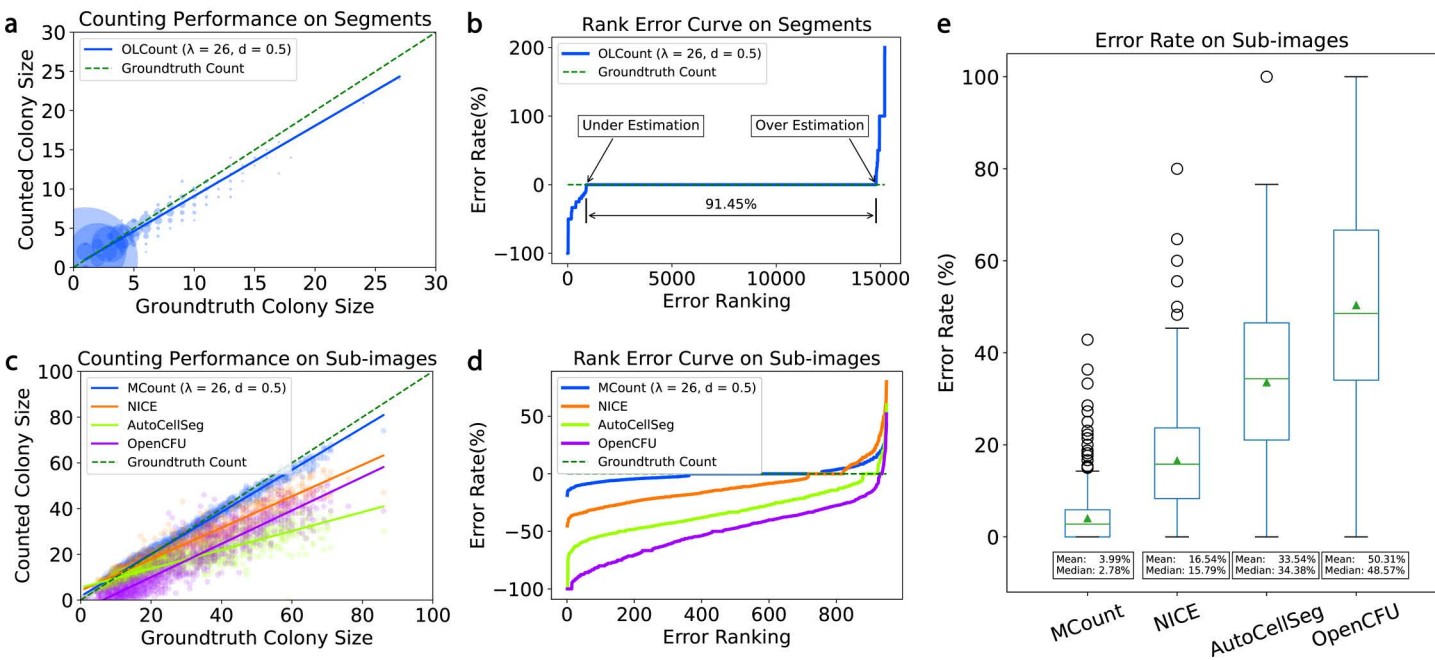

**Fig 4. Comparison of MCount ( $\lambda = 26$ and $d = 0.5$ ) and NICE performance in recognizing colony segments and sub-images. (a)** Bubble plot of MCount counting result versus ground truth label on 15,847 colony segments. The size of each bubble at location (x,y) is proportional to the number of segments that have x merged colonies but are recognized as y colonies by MCount. The blue line represents the regression line for MCount, while the green dashed line represents the regression line for a perfect algorithm that always gives the result as the label. MCount exhibits slight underestimation as the blue line is lower than the dashed green line. **(b)** Rank error curve of error rate values for 15,847 segments. MCount recognizes 91.45% of segments with zero error rate. **(c)** Bubble plot of Mcount, NICE, AutoCellSeg, and OpenCFU counting results versus ground truth count label on 960 colony sub-images. The blue line is much closer to the dashed green line compared to the orange line, indicating that MCount is more accurate in recognizing sub-images. **(d)** Comparison of MCount, NICE, AutoCellSeg, and OpenCFU performance in underestimation or overestimation of colony numbers. MCount has fewer underestimated results and less severe underestimation and overestimation compared to NICE. **(e)** Boxplot of error rate on 960 sub-images for MCount and NICE. MCount has a much lower average error rate and a smaller interquartile range, indicating more consistent performance.

MCount provides a good estimate for colony recognition on the segment dataset. It is worth noting that the bubbles representing coordinates (1,1), (2,2), and (3,3) are large compared to other bubbles because segments with 1-, 2-, and 3-colonies account for a large portion (59.6%, 18.52%, and 7.93%, respectively) of the segment dataset, and most of them are correctly evaluated by MCount. Therefore, the regression line of MCount largely depends on segments with few colonies, even though the segment size in the dataset ranges from 1 to 27. MCount tends to slightly underestimate the number of colonies, but it is still accurate for most segments, as 92.26% of segments have no more than 5 colonies. In this case, the actual bias resulting from underestimation is zero because the counted number can only be natural numbers. The counted number monotonically decreases as $\lambda$ increases, as shown in S4 Fig (a), and this underestimation issue can be addressed by choosing an appropriate value for $\lambda$. To better visualize the accuracy of MCount on colony segments, we used a rank error curve in Fig 4b, where all 15,847 segments are ranked according to their error rate. The x-axis represents the rank of each segment, and the y-axis represents the corresponding error rate. When the error rate is larger than zero, counting tends to overestimate, while counting underestimates when the error rate is smaller than zero. A zero error rate was achieved on 91.45% of the segments, demonstrating MCount's high accuracy on colony segments.

The goal of the colony counting algorithm is to give an accurate count on sub-images rather than segments, therefore we further evaluated the accuracy of MCount on sub-images (Fig 4c-d) and compared it with NICE, AutoCellSeg, and OpenCFU (Fig 4e). Notably, many algorithms either lack accessible source code [17], are unavailable for download [23], or fail to install correctly [8,18]. Consequently, we selected the most popular and widely used tools — NICE [9], OpenCFU [10], AutoCellSeg [11] — for comparison.

Fig 4c is a bubble plot showing all algorithms versus the ground-truth label on sub-images. All algorithms have relatively uniform distributions indicated by the uniform bubble sizes. While MCount only slightly underestimates the number of colonies, which can be addressed by tuning the $\lambda$ value as shown in S4 Fig (b), NICE, AutoCellSeg, and OpenCFU largely underestimate the number of colonies. Since AutoCellSeg tends to count every continuous region as a single colony regardless of its shape, it has the most severe underestimation issue; as OpenCFU struggles with low-resolution images, its fitted line cannot even penetrate the origin (x,y) = (0,0). Fig 4d shows the rank error curve of all algorithms, with MCount having better accuracy than the others. Fig 4e shows the boxplot of both algorithms' recognition error on each of the 960 sub-images, with MCount having a 3.99% average error rate, substantially lower than the average error rates of NICE/ AutoCellSeg/ OpenCFU of 16.54%/ 33.54%/ 50.31% respectively.

To gain insights into the sensitivity of MCount to hyperparameters, we explored two key questions. First, we examined whether the optimal hyperparameters for each sub-dataset significantly differed from the global optimal parameters, noting that these 10 sub-datasets were generated under varying experimental conditions. The results, as shown in S1 Table, indicate that the variation in optimal hyperparameters across different sub-datasets is minimal, and their corresponding recognition error rates are all lower than the error rate observed when using the global optimal hyperparameters (3.99%). Additionally, we observed a pattern where the value of the hyperparameter $d$ tends to increase as the mean colony number decreases. When culture time is consistent, sparser colony density typically leads to larger colony size, suggesting that $d$ is positively related to colony size. The second question we explored was how well the globally determined optimal hyperparameters perform across different sub-datasets. The results, presented in S2 Table, show that the globally optimized hyperparameters applied to these 10 groups allow MCount to maintain robust performance across varied conditions, with average error rates that remain significantly lower than NICE.

The processing time of MCount was evaluated on the dataset by randomly selecting 100 sub-images. The average time per sub-image was found to be less than 1.7 seconds, indicating that MCount can meet the demand for high-throughput processing. However, the processing time can be further optimized by using a better optimizer solver. A comparison of the processing time for MCount and other solutions is presented in S3 Table.

### 3.3. Hyperparameter optimization

In previous sections, we utilized a standard method of grid search and cross-validation with the goal of minimizing the average error rate on the entire dataset to tune the hyperparameter set $(d, \lambda)$, which can be further improved by using techniques such as random search [40]. However, this can be computationally intensive and time-consuming when exploring the hyperparameter space in practice. Furthermore, this method requires a pre-labeled dataset with a large quantity of data points, e.g., 960 (as shown earlier), which is impractical when deploying MCount for other counting tasks. In this section, we explore alternative methods for determining the hyperparameter set $(d, \lambda)$ that only require a few labeled or even unlabeled data points. Table 1 summarizes the hyperparameter optimization methods used in this work.

Although $d$ and $\lambda$ are not completely independent, they can be considered separately since they control different aspects of the algorithm. Specifically, $d$ governs the level of detail in the contour representation, while $\lambda$ controls the number of circles used in the optimization step. In this way, we can decouple these two hyperparameters and investigate them separately.

We propose an empirical method to determine the appropriate value of $d$. MCount can recognize elliptical segments that are usually comprised of two colonies by dividing the contour of the elliptical colonies into multiple contour segments, as shown in Fig 3a. We began with $d = 100$ and selected a few elliptical segments that appeared to have two colonies. We then visualized the contour of each selected elliptical segment and gradually decreased the value of $d$ until the contour was divided into $3 \sim 10$ segments. By applying this approach to the benchmark, we determined that $d = 0.5$ is a suitable value. In the following sections, we focus on statistical methods for tuning $\lambda$ and examine their consistency using statistical procedures.

**3.3.1. The selection of $\lambda$ by averaging a small number of labeled images.** Although labeling hundreds of colony images to calibrate MCount requires extensive effort, for other counting tasks labeling only a few dozen images may be feasible. Therefore, we are interested in statistical methods that require only a few labeled images to tune $\lambda$. Here we propose one such method:

(1) Obtain a small number of labeled images (e.g., $n = 10$).

(2) For each labeled image, find the value of $\lambda_i$ such that MCount gives the correct colony count. Note that the count monotonically decreases as $\lambda$ increases, making it easy to find $\lambda_i$.

(3) Set the hyperparameter $\lambda$ to the average value, $\lambda = \bar{\lambda} = \frac{1}{n}\sum_i \lambda_i$.

**Table 1. Summary of hyperparameter optimization methods used in MCount.** The table includes the method number, image type (labeled or unlabeled), number of images used for optimization, hyperparameter $d$ optimization method, and hyperparameter $\lambda$ optimization method. Method 1 uses grid search with cross-validation on 960 labeled images (Section 3.2 & 3.3). Method 2 and method 3 decoupled the optimization of $d$ and $\lambda$ and use the same empirical method for determining $d$ (Section 3.3). Method 2 uses the average $\lambda$ of 10 ~ 20 labeled images (Section 3.3.1), while method 3 chooses the value of $\lambda$ that leads to equidispersion on 40 ~ 50 unlabeled images (Section 3.3.2).

| | Image type | Number of images | $d$ optimization | $\lambda$ optimization | Section |
|---|---|---|---|---|---|
| 1 | Labeled | 960 | Grid Search + Cross Validation | | |
| 2 | Labeled | 10 ~ 20 | Empirical | Average of Samples | 3.3.1 |
| 3 | Unlabeled | 40 ~ 50 | Empirical | Equidispersion Assumption | 3.3.2 |

In order to examine whether the proposed method can consistently lead to an appropriate $\lambda$ value and determine how many images are needed in step (1), we carried out a simulation using the benchmark dataset of 960 labeled images. The simulation procedure is as follows:

(1) Randomly sample $n$ images from the 960 labeled images.

(2) For each labeled image, find the value of $\lambda_i$ such that MCount gives the same colony count as the label. Note that the count monotonically decreases as $\lambda$ increases, making it easy to find $\lambda_i$.

(3) Calculate the average recognition error rate on the 960 images with $\lambda = \bar{\lambda} = \frac{1}{n}\sum_i \lambda_i$.

(4) Repeat steps (1) to (3) 1000 times.

The simulation procedure allows us to investigate the distribution of the average recognition error rates from 1000 trials to see whether this hyperparameter-tuning method can consistently lead to low error rates. Fig 5 is a boxplot of simulation results under different values of $n$. Not surprisingly, a larger $n$ leads to a narrower distribution and a lower mean/median. When $n = 10\ (20)$, 985 (998) out of 1000 trials have a $\lambda$ value leading to a low error rate ($<8\%$) on the 960 images, which indicates consistently low recognition error rates. In practice, labeling ten to twenty colony images is a common practice in bio-labs, making it easy to calibrate MCount when applied to future counting tasks.

**3.3.2. The selection of $\lambda$ based on equidispersion assumption on unlabeled images.** In many labs, there are often cases where acquiring a large number of images is feasible, but labeling even a single image is challenging. These situations require statistical methods that

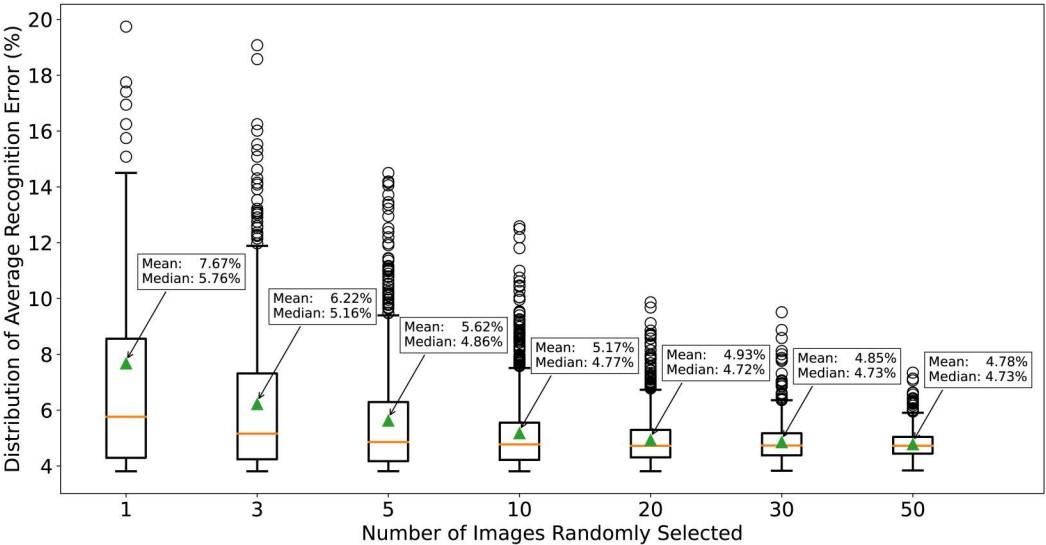

**Fig 5. Distribution of average recognition error rates on the benchmark dataset when repeatedly implementing the hyperparameter-tuning method using $n$ labeled images 1000 times.** In each replicate, a few (1, 3, 5, 10, 20, 30, or 50) images are randomly selected from the benchmark dataset, and $\lambda$ is tuned for each image to a proper value so that MCount gives the same counting number as the label. Then, the average of $\lambda$ is chosen for this replicate, and the average recognition error rate is calculated on the benchmark using this $\lambda$. By simulating this procedure for 1000 replicates, we can plot the distribution of average recognition error rates. As expected, increasing $n$ results in a narrower distribution of average recognition error rates, leading to more consistent performance. Note that when $n = 10$, all recognition errors fall in the range of 3.5% to 13% with a mean of 5.17% (median of 4.77%), much lower than the recognition error rate of NICE at 16.54% (15.79%).

can tune $\lambda$ using unlabeled images. Further, introducing some prior information could help make good use of the unlabeled data. One such prior is an assumption that the distribution of colony numbers features equidispersion, meaning that the mean is equal to the variance, a key property of the Poisson distribution. We discuss this prior information in more detail before introducing and examining our method in the following section. By utilizing this prior information/assumption, we hypothesize that we can obtain a reasonably accurate $\lambda$ value using unlabeled images. By analyzing the simulation results we can determine the minimum number of unlabeled images required to obtain an accurate $\lambda$ value for a specific task.

**3.3.2.1. Hypothesis test for assessing equidispersion:** It is assumed that many biological phenomena exhibit a Poisson distribution, and colony numbers are an example, where a fixed volume of liquid is independently and randomly sampled from the same source for every plating [47,48]. In this case, the equidispersion property is naturally satisfied.

However, it is important to note that the prior knowledge chosen in this method is the equidispersion property, not specifically the Poisson distribution. Equidispersion is a weaker requirement than the Poisson distribution, and it is possible that other distributions can also exhibit equidispersion. In S2 File and S4 Table, we further discuss this scenario and show that our benchmark is closer to a normal distribution than a Poisson distribution, although our equidispersion assumption still leads to a low recognition error.

We prefer assuming equidispersion to the Poisson distribution because it can be challenging to use statistical methods to confirm that a distribution follows Poisson, while equidispersion can be tested more easily. Here we propose a simple hypothesis test to assess equidispersion, known as the Poisson Dispersion test, which also serves as a likelihood test for the Poisson distribution:

$$H_0 : \text{the data has equidispersion, } i.e., \; mean = variance$$

$$H_1 : \; mean \neq variance$$

The test statistic is:

$$D = \sum_{N}^{i=1} \frac{\left(X_i - \bar{X}\right)^2}{\bar{X}} \tag{5}$$

Then we can calculate the p-value associated with the chi-square distribution (N-1 degrees of freedom). If the p-value is greater than the chosen level of significance, we fail to reject the null hypothesis $H_0$ and can assume that equidispersion is satisfied.

**3.3.2.2. Proposed method and its examination:** Assuming that equidispersion is satisfied, we propose a statistical method that utilizes unlabeled images to tune $\lambda$ as follows:

(1) Obtain a set of unlabeled images (e.g., $n = 40$ ).

(2) Apply MCount to the images and obtain the counting results under different $\lambda$ values.

(3) Calculate the mean and variance of the counting results under different $\lambda$ values and find a value $\hat{\lambda}$ where the mean equals the variance.

(4) Set the hyperparameter $\lambda = \hat{\lambda}$ .

To evaluate the effectiveness of the proposed method using unlabeled images, we conducted a simulation procedure as follows:

(1) Randomly select $n$ images from a dataset comprising 96 labeled images with equidispersion. Erase the label of the selected images.

(2) Apply MCount to the $n$ images and obtain counting results under different $\lambda$ values.

(3) Calculate the mean and variance of the counting results under different $\lambda$ values and find a value $\hat{\lambda}$ where the mean equals the variance.

(4) Calculate the average recognition error rate on 96 labeled images with $\lambda = \hat{\lambda}$.

(5) Repeat steps (1) to (4) 1000 times.

The boxplot in Fig 6 shows the distribution of the 1000 average recognition error rates for different values of $n$. Like the case for labeled images, larger $n$ values lead to a narrower distribution and smaller mean. Interestingly, the median remains constant regardless of $n$. When $n \geq 40$, the maximum line of the boxplot is lower than 8%, indicating consistently low recognition error rates. However, the required $n$ value is much larger than that for labeled images, as unlabeled images contain less information, requiring more training data for hyperparameter optimization.

Notably, the equidispersion criterion helps identify a range of reasonable lambda values, though it may not always yield a unique solution. In practice, we often observe that within a reasonable range—typically near the default values—a single optimal lambda value emerges. This is because variance is generally less sensitive to changes in lambda, while the mean is more responsive, aiding in narrowing down the solution. When multiple lambda values meet the criterion, they are applied to the colony counting process and manually inspected to determine the most accurate outcome, with clear indications when overestimation or underestimation occurs.

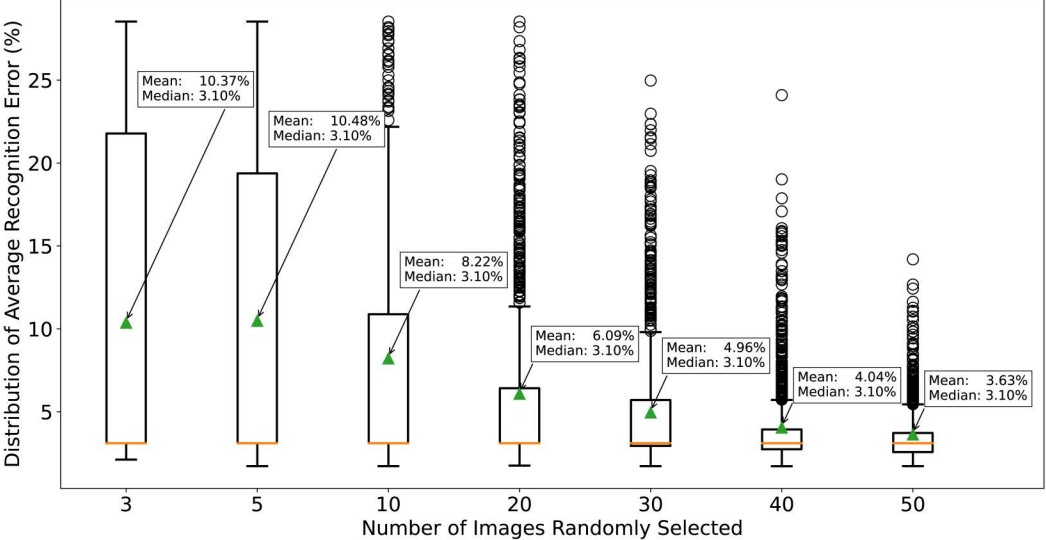

**Fig 6. Distribution of average recognition error rates on 96 images with equidispersion when repeatedly implementing the hyperparameter-tuning method using $n$ unlabeled images 1000 times.** In each replicate, a few (3, 5, 10, 20, 30, 40, or 50) images with the label erased are randomly selected from 96 images, and $\lambda$ is tuned so that the mean of the counting equals the variance. Then, the average recognition error rate is calculated on the 96 labeled images using this $\lambda$. By simulating this procedure for 1000 replicates, we can plot the distribution of average recognition error rates. As $n$ increases, the distribution of average recognition error rates becomes narrower, leading to a more consistent distribution. Note that only when $n > 40$, the maximum line of the box plot is lower than 8%, and the mean is equal or lower than 4.04%. The required $n$ is much larger than the case for labeled images because unlabeled images have less information, requiring more training data for hyperparameter optimization. These results demonstrate that the proposed method is effective in achieving consistently low error rates using only a small number of unlabeled images for hyperparameter tuning.

## 4. Conclusion

In this work, we propose a colony counting solution, MCount, that can recognize merged colonies that frequently occur in high-throughput workflows, which is beyond the capabilities of current solutions that adopt region-based algorithms. MCount extracts contour information and combines it with regional information using an optimization algorithm. To evaluate the performance of MCount, we prepared a GFP-fluorescent *E. coli* DH10-beta-based colony dataset, which is comprised of both sub-images and labeled segments. MCount maintains an average recognition error of 3.99% on the sub-image dataset (using grid search and 10-fold cross-validation to minimize the average error rate), which is much lower than current solutions like NICE of 16.54%.

Given that MCount only has two hyperparameters, it will be easy to deploy for other counting tasks. In addition to the standard hyperparameter optimization method, which requires several labeled images, we also proposed two methods that require a small number of labeled images or unlabeled images, respectively. To examine the statistical properties of the proposed methods, we conducted simulations and found that they all guarantee consistently low error rates compared to existing methods. The simulations showed that the method requiring labeled images achieved low error rates with as few as 10 labeled images, while the method suitable for unlabeled images required at least 40 images to achieve consistently low error rates.

Overall, the statistical evaluation of the proposed methods provides a strong basis for their potential deployment in various counting tasks. Future applications of this solution include colony classification for multiple strains on the same plate, which could be achieved by classification algorithms such as K-nearest neighbor based on colony color and size.

## Supporting Information

**S1 Fig. The Polygon Approximation Algorithm uses turning points on the contour to represent an inscribed polygon, where d controls how many turning points are generated.** **(a)** For consecutive three turning points $T_{i-1}$, $T_i$, and $T_{i+1}$, $d$ represents the distance from $T_i$ to the line $\overline{T_{i-1}T_{i+1}}$. **(b)** The larger the value of $d$, the more turning points are generated to represent the contour. The turning points are represented as black dots, while contour segments divided by turning points are shown in different colors.
(TIF)

**S2 Fig. A dataset that includes colony segments, labeled using historical photographs, is used to optimize and evaluate the performance of MCount. (a)** 960 sub-images are obtained by cropping 10 plates of fluorescent *E. coli* NEB10-beta and further divided into 15,847 segments using foreground extraction and segmentation. **(b)** The colony number distribution of sub-images shows that most sub-images have 10 ~ 40 colonies. **(c)** The percentage of single-colony, two-colony, and three-colony segments is 59.6%, 18.52%, and 7.93%, respectively, which takes 86.05% in total. A well-performing algorithm is expected to correctly recognize almost all none and mildly merged colonies. The remaining percentage of segments, merged in a denser manner, requires the algorithm to infer sophisticated shapes. Note that 3.90% of segments are invalid because the merging of colonies is too severe to be labeled correctly, denoted as -1 in the left figure. **(d)** All segments are labeled according to their shape in a photograph taken about 4 hours ago to ensure labeling accuracy, including segments that are hard for humans to label.
(TIF)

**S3 Fig.** The optimal values of $d$ and $\lambda$ are determined using a different definition of recognition error. **(a)** The average error rate on the segment dataset is plotted against $d$ when $\lambda = 26$. The minimum average error rate of 8.50% is achieved at $d = 0.5$. **(b)** The average error rate on the segment dataset is plotted against $\lambda$ when $d = 0.5$. The minimum average error rate of 8.46% is achieved at $\lambda = 28$. An error of each segment is defined as $1(MCount \neq Label)$ and the error was averaged across all segments, denoting the average error rate.
(TIF)

**S4 Fig.** Increasing $\lambda$ results in monotonously less counting leading to underestimation of colony number. The plot shows the MCount counting result with different $\lambda$ values versus ground truth label on **(a)** 15,847 colony segments and **(b)** 960 sub-images, respectively. The green dashed line represents the regression line for a perfect colony estimator that always gives the result as the label, while the colored line represents the regression line for MCount. By tuning $\lambda$, it is possible to address the overestimation/underestimation issue. Increasing $\lambda$ results in a monotonous decrease in the number of colonies counted, leading to an underestimation of colony number.
(TIF)

**S1 Table. Variation of optimal hyperparameters across 10 sub-datasets generated under different experimental conditions.** Mean and variance of colony number, optimal hyperparameter values, and the corresponding recognition error rate for each sub-dataset. The results demonstrate minimal variation in hyperparameters across sub-datasets and consistently lower error rates compared to the global settings (3.99%).
(DOCX)

**S2 Table.** Performance of MCount using globally optimized hyperparameters ($\lambda = 26$ and $d = 0.5$) across 10 sub-datasets. The table presents the average error rates for each sub-dataset when applying the global optimal hyperparameters. The results indicate that MCount maintains robust performance across varied experimental conditions, with error rates significantly lower than those achieved by NICE.
(DOCX)

**S3 Table. Processing time of different colony counting solutions for 100 randomly selected sub-images.** The optimization solver for MCount is 'COIN_CMD' in PuLP, and the solver can be changed to achieve faster processing times at the cost of sacrificing accuracy.
(DOCX)

**S4 Table. Mean, variance, and p-values of tests for 10 datasets.** The mean and variance values were calculated from the 96 colony counts of each dataset. The first column represents the dataset index. The second column represents the mean value of the 96 colony counts, and the third column represents the variance of the 96 colony counts. The 4th and 5th columns show the p-values for Kolmogorov–Smirnov (KS) tests for Normal and Poisson distributions, respectively, under the significant level of 0.05, where no rejection is made for any datasets. The 6th column shows the p-values for Poisson Dispersion tests, and datasets whose null hypotheses are rejected under the significant level of 0.05 are denoted with a star symbol *.
(DOCX)

**S1 File. Preparation and discussion on bacterial colony dataset.** Discussion, cultivation and preparation on *E. coli* Dataset used in this study.
(DOCX)

**S2 File. Discussion on the distribution of colony numbers.**

(DOCX)

## Author contributions

**Conceptualization:** Sijie Chen.

**Data curation:** Sijie Chen, Po-Hsun Huang, Yuhe Cui.

**Formal analysis:** Sijie Chen.

**Funding acquisition:** Cullen R. Buie.

**Investigation:** Sijie Chen.

**Methodology:** Sijie Chen, Hyungseok Kim.

**Project administration:** Po-Hsun Huang.

**Software:** Sijie Chen.

**Supervision:** Cullen R. Buie.

**Validation:** Sijie Chen.

**Writing – original draft:** Sijie Chen, Hyungseok Kim, Cullen R. Buie.

**Writing – review & editing:** Sijie Chen, Hyungseok Kim, Cullen R. Buie.

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
