## [Decision Letter · Decision Letter 0]

11 Mar 2024

PONE-D-24-05814MCount: An automated colony counting tool for high-throughput microbiologyPLOS ONE

Dear Dr. Buie,

Thank you for submitting your manuscript to PLOS ONE. After careful consideration, we feel that it has merit but does not fully meet PLOS ONE’s publication criteria as it currently stands. Therefore, we invite you to submit a revised version of the manuscript that addresses the points raised during the review process.

We look forward to receiving your revised manuscript.

Kind regards,

Florian Rehfeldt

Academic Editor

PLOS ONE

Journal Requirements:

"I have read the journal's policy and the authors of this manuscript have the following competing interests: C.R.B. is a Co-Founder and Advisor of Kytopen Corp.  The remaining authors declare no competing interests."

4. Thank you for uploading your study's underlying data set. Unfortunately, the repository you have noted in your Data Availability statement does not qualify as an acceptable data repository according to PLOS's standards.

Reviewers' comments:

Reviewer's Responses to Questions

**Comments to the Author**

1. Is the manuscript technically sound, and do the data support the conclusions?

Reviewer #1: Yes

Reviewer #2: Yes

2. Has the statistical analysis been performed appropriately and rigorously? 

Reviewer #1: Yes

Reviewer #2: Yes

3. Have the authors made all data underlying the findings in their manuscript fully available?

Reviewer #1: Yes

Reviewer #2: Yes

4. Is the manuscript presented in an intelligible fashion and written in standard English?

Reviewer #1: Yes

Reviewer #2: Yes

5. Review Comments to the Author

Reviewer #1: In their manuscript, Chen et al. present a sophisticated workflow for automated colony counting, which they name 'MCount'. The manuscript is well written, and in principle suitable for publication. Before publication, the authors may want to consider the following points:

1.) In the introduction, the authors discuss several existing tools for colony counting (NICE, AutoCellSeg, OpenCFU), but in the results section they only compare the performance of MCount to NICE. Since the authors went through the effort of creating a ground-truth dataset, it would be nice to see also the performance of the other methods on this dataset.

2.) MCount has two major tunable hyperparameters. As I understood, this is not the case for NICE. I acknowledge the effort the authors have put into guiding users in how to determine those hyperparameters. Still, to understand to what extent the improved performance of MCount is due to those tuned hyperparameters, it would be interesting to see how MCount would perform 'out-of-the-box' without parameter tuning, and how this would compare to NICE. To assess this, the authors could test how optimized hyperparameters tuned on the current dataset perform on an independent dataset, ideally obtained in an independent setting (e.g. by another lab).

3.) Along those lines, how sensitive is the optimum of the hyperparameters to various datasets? Especially datasets with similar colony density but obtained with a different imaging setup?

4.) To determine the hyperparameter lambda from unlabelled data, the authors present a strategy building on a equidispersion criteria. I did not understand, whether this criteria leads to a unique optimum for lambda? If not, what are the implications for this approach?

5.) The authors apply their method using a fluorescent strain. Is fluorescence imaging required for the good performance?

Reviewer #2: This work presents a classical image analysis workflow for counting CFUs. The method is described in all necessary detail, code and data are available, and the results are quantitatively analyzed. The main limitation of this manuscript is that it lacks a thorough comparison to existing literature and state-of-the-art. Apart from some qualitative comparison in the first figure and a few plots comparing their work to NICE in Fig. 4, no such comparison is performed. The literature cited is mostly from computer vision applications to other types of overlapping objects, but not microbiology or bioimaging.

The problem of counting objects is nowadays usually solved with object detection and/or segmentation neural networks, some of which are specifically taylored towards separating overlapping convex objects such as cell nuclei (e.g. Stardist, Cellpose). In the field of colony counting, some classical and ML-based literature not cited here (just as examples) include:

https://doi.org/10.2144%2F000112018

https://www.nature.com/articles/s41598-020-72979-4

https://link.springer.com/doi/10.1007/s10796-009-9149-0

http://arxiv.org/abs/2009.00926

https://doi.org/10.1364/OSAC.396603

https://doi.org/10.1016/j.tim.2021.01.006

https://vciba.springeropen.com/articles/10.1186/s42492-022-00122-3

The literature is for example reviewed in

https://doi.org/10.1016/j.tim.2021.01.006

https://link.springer.com/article/10.1007/s10462-021-10082-4

It is not necessary to cite all of this work in the manuscript, but the authors are encouraged to take the effort and spend some time studying the current literature, and then add a discussion of their work with respect to the current state of the art, and ideally also some more quantitative comparison.

Some technical remarks: it would be great if more documentation, explanation of the individual steps, and installation instructions could be added to the Jupter Notebooks. The figures appear in very low quality in the PDF, but this might be a production issue unrelated to the submitted material.

6. PLOS authors have the option to publish the peer review history of their article (what does this mean? ). If published, this will include your full peer review and any attached files.

**Do you want your identity to be public for this peer review?** For information about this choice, including consent withdrawal, please see our Privacy Policy .

Reviewer #1: No

Reviewer #2: No

---

## [Author Response · Author response to Decision Letter 0]

10 Sep 2024

Response to Editor

[Comment]

Thank you for uploading your study's underlying data set. Unfortunately, the repository you have noted in your Data Availability statement does not qualify as an acceptable data repository according to PLOS's standards. At this time, please upload the minimal data set necessary to replicate your study's findings to a stable, public repository (such as figshare or Dryad) and provide us with the relevant URLs, DOIs, or accession numbers that may be used to access these data. For a list of recommended repositories and additional information on PLOS standards for data deposition, please see https://journals.plos.org/plosone/s/recommended-repositories.

[Response]

We appreciate the editor’s time reviewing our manuscript and Data Availability statement. We accept the suggestion and have uploaded the data set to Dryad. The relevant URL or DOI to access the data set is as follows: https://doi.org/10.5061/dryad.2280gb62f (forthcoming as of September 5, 2024; for temporary access in the meantime, please access via https://datadryad.org/stash/share/xh0Ek1hMtE-lihJ5wB590g-rnLcfL6G0irdWlaVxjhE). We have added the URL to the section “Data Availability” as follows:

“All methods implemented and data used are publicly available on Dryad (https://doi.org/10.5061/dryad.2280gb62).”

Response to Reviewer 1

[Comment 1-1]

In the introduction, the authors discuss several existing tools for colony counting (NICE, AutoCellSeg, OpenCFU), but in the results section they only compare the performance of MCount to NICE. Since the authors went through the effort of creating a ground-truth dataset, it would be nice to see also the performance of the other methods on this dataset.

[Response 1-1]

We appreciate the reviewer’s valuable comment. Indeed, we previously did not compare the results of MCount to AutoCellSeg and OpenCFU and instead outlined their limitations as described in Introduction and in Figure 1. Following reviewer’s comment, we have performed colony counting using AutoCellSeg and OpenCFU and have now added their results to Figure 4 for comparison.

As expected, we see that MCount has shown the lowest error rate compared to other methods such as NICE, AutoCellSeg, and OpenCFU. We have added our motivations and observations of the comparison study (lines 305-309 of main text)

[Comment 1-2]

MCount has two major tunable hyperparameters. As I understood, this is not the case for NICE. I acknowledge the effort the authors have put into guiding users in how to determine those hyperparameters. Still, to understand to what extent the improved performance of MCount is due to those tuned hyperparameters, it would be interesting to see how MCount would perform 'out-of-the-box' without parameter tuning, and how this would compare to NICE. To assess this, the authors could test how optimized hyperparameters tuned on the current dataset perform on an independent dataset, ideally obtained in an independent setting (e.g. by another lab)

[Response 1-2]

We appreciate your concerns on hyperparameter tuning and would like to address them as follows.

NICE Also Has Hyperparameters: It's important to note that NICE, like MCount, includes hyperparameters, such as sigma, i.e. the threshold of Otsu algorithm used in NICE. While many users might default to the standard values, these parameters significantly impact NICE's performance. However, in practice, many biologists prioritize experimental efficiency or believe that the error introduced by hyperparameters is smaller than the inherent variability in biological experiments. As a result, the adjustment process in NICE often involves providing a single example image and manually tuning the parameters based on subjective judgment, and in many cases, researchers simply use the default values without further calibration.

Necessity of Hyperparameters: We believe that having tunable hyperparameters is essential for a tool like MCount, as it allows the tool to be more adaptable to different scenarios and achieve optimal performance. From a data science perspective, the best approach to obtain optimal hyperparameters is to construct a small, well-labeled dataset that reflects the typical experimental setting, which can then be used to calibrate values like sigma. This approach ensures that MCount can maintain high accuracy across various experimental conditions and image qualities. However, the choice of hyperparameters presents a dilemma: on one hand, we want them to be sensitive enough to ensure adaptability across different experimental conditions; on the other hand, we want them to be less sensitive, making the tool easier to use for those who prefer simplicity. Given that MCount has the more advanced capability to differentiate overlapping colonies, its parameters are necessarily more sensitive compared to NICE.

Performance on Different Datasets: While we did not conduct experiments on a completely new and independent dataset, as our dataset, which consists of 10 groups of 96 images each, already exhibits sufficient variance. These 10 groups vary in terms of colony size, density, and even the setup used during image collection (e.g., variations in camera positioning, background lighting, etc.). This natural diversity in the dataset makes it suitable for evaluating the robustness of MCount's performance. We added corresponding clarification in the main text (lines 213-216). To address the reviewer's concerns, we applied the optimized hyperparameters to these 10 groups and assessed MCount's accuracy. Below is a summary of the results, which has now been added into Supplemental Information as S2 Table. These results demonstrate that MCount maintains strong performance across varied conditions within the original dataset, with an average error rate that remains significantly lower than NICE (16.54%). This suggests that the optimized hyperparameters are robust and generalizable across different experimental settings.

[Comment 1-3]

Along those lines, how sensitive is the optimum of the hyperparameters to various datasets? Especially datasets with similar colony density but obtained with a different imaging setup?

[Response 1-3]

The sensitivity of the optimal hyperparameters can indeed be characterized from two perspectives:

Effect of Hyperparameter Adjustment on Results: The first perspective is to observe how adjusting the hyperparameters affects the results. We previously addressed this and presented the result in S3 Fig, which illustrates how variations in hyperparameters impact the performance of MCount.

Variation of Optimum Across Different Datasets: The second perspective, as the reviewer suggested, concerns whether the optimum hyperparameters differ across datasets, particularly those with similar colony density but obtained using different imaging setups. To explore this, we calculated the optimal hyperparameters for each of the 10 sub-datasets in our study. Again, these 10 sets vary in terms of colony size, density, and the imaging setup. The results have now been summarized in S1 Table as follows:

These results indicate that while there is some variation in the optimal hyperparameters across different sub-datasets, MCount maintains robust performance. The differences in optimal hyperparameters are likely influenced by factors such as colony size, density, and variations in the imaging setup (e.g., camera positioning, background lighting). This analysis provides insights into the robustness and adaptability of MCount across different experimental conditions.

Additionally, we observed a pattern where the value of the hyperparameter 𝑑 tends to increase when the mean colony number decreases (considering a consistent culture time, which results in larger colonies). This correlation suggests that 𝑑 is positively related to colony size, which aligns with our expectations.

It's also worth noting that the recognition error rates for these individually optimized sub-datasets are significantly lower than the error rate observed when using a single set of optimal parameters for all 10 datasets combined (4%). This further underscores the benefit of tuning hyperparameters to suit specific experimental conditions.

Our assessment of MCount across different datasets has now been added to the main text (lines 320-331).

[Comment 1-4]

To determine the hyperparameter lambda from unlabelled data, the authors present a strategy building on a equidispersion criteria. I did not understand, whether this criteria leads to a unique optimum for lambda? If not, what are the implications for this approach?

[Response 1-4]

As the reviewer rightly pointed out, the equidispersion criterion does not guarantee a unique solution for the lambda parameter. Theoretically, there could be multiple lambda values that satisfy this criterion. However, this does not render the equidispersion criterion ineffective; rather, it serves as a tool to identify a range of lambda values that are relatively reasonable. We have added corresponding clarifications to the main text (lines 467-473).

Practical Observations: In our experimental experience with various datasets, we typically observe that within a reasonable range (usually around the default values of the parameters), we tend to obtain a single solution for lambda. This is because variance is relatively insensitive to changes in lambda, while the mean is more responsive to variations in lambda, helping to narrow down the solution.

Handling Multiple Lambda Values: In cases where multiple lambda values meet the equidispersion criterion, we can use these values to perform the colony counting and then manually inspect the results to determine which value provides the most reasonable outcomes. We have encountered situations where certain lambda values resulted in clearly overestimated or underestimated colony counts, making it evident which lambda should be chosen.

[Comment 1-5]

The authors apply their method using a fluorescent strain. Is fluorescence imaging required for the good performance?

[Response 1-5]

While fluorescence imaging can enhance performance, it is not strictly required. We have also successfully applied our method to non-fluorescent images in other, unpublished research. For instance, we used our approach to count colonies with different morphological characteristics in images containing mixtures of colonies with three different colors or sizes. These experiments demonstrate that our method is versatile and can effectively adapt to non-fluorescent imaging scenarios, provided the images are of sufficient quality to distinguish between colonies.

In this study, fluorescence imaging was primarily used to facilitate the manual annotation of the dataset. Fluorescence provides clear and distinct colony boundaries, making it easier to create accurate ground-truth labels, which are essential for training and validating the model.

We have added corresponding edits to the main text (lines 209-212).

Response to Reviewer 2

[Comment 2-1]

This work presents a classical image analysis workflow for counting CFUs. The method is described in all necessary detail, code and data are available, and the results are quantitatively analyzed. The main limitation of this manuscript is that it lacks a thorough comparison to existing literature and state-of-the-art. Apart from some qualitative comparison in the first figure and a few plots comparing their work to NICE in Fig. 4, no such comparison is performed. The literature cited is mostly from computer vision applications to other types of overlapping objects, but not microbiology or bioimaging. The problem of counting objects is nowadays usually solved with object detection and/or segmentation neural networks, some of which are specifically taylored towards separating overlapping convex objects such as cell nuclei (e.g. Stardist, Cellpose). In the field of colony counting, some classical and ML-based literature not cited here (just as examples) include:

https://doi.org/10.2144%2F000112018

https://www.nature.com/articles/s41598-020-72979-4

https://link.springer.com/doi/10.1007/s10796-009-9149-0

http://arxiv.org/abs/2009.00926

https://doi.org/10.1364/OSAC.396603

https://doi.org/10.1016/j.tim.2021.01.006

https://vciba.springeropen.com/articles/10.1186/s42492-022-00122-3

The literature is for example reviewed in

https://doi.org/10.1016/j.tim.2021.01.006

https://link.springer.com/article/10.1007/s10462-021-10082-4

It is not necessary to cite all of this work in the manuscript, but the authors are encouraged to take the effort and spend some time studying the current literature, and then add a discussion of their work with respect to the current state of the art, and ideally also some more quantitative comparison.

[Response 2-1]

We appreciate the reviewer for pointing out this limitation in our manuscript. We have taken the following steps to address the comment.

As per the reviewer’s suggestion, we have incorporated additional references to bioimaging and neural network (or deep learning) related literature in the manuscript. We also provided a paragraph discussing on why we chose not to adopt neural network (or deep learning) approaches. We would like to clarify that, broadly speaking, our method can also be classified as a machine learning (ML) method, which is why we avoid labeling other approaches as exclusively "ML methods." Perhaps it would be more appropriate to refer to our approach as a "classical method" and to the methods the reviewer mentioned as "neural network methods." Changes were made in throughout the Introduction of main text, more specifically at lines 79-87.

We have made an effort to include more quantitative comparisons in our revised manuscript. Specifically, we have modified Figure 4 to further include the counting performance of OpenCFU and AutoCellSeg. We believe this comparison is reasonable because, as the reviewer noted, one of the 2022 ML papers referred also uses OpenCFU and AutoCellSeg as benchmarks. We also attempted to include comparisons with newer tools, such as the ones referred above. However, we encountered several challenges: some of these tools do not provide source code, making replication difficult; others offer software that we were unable to download, or that did not function correctly after download. Please see the details of these challenges in the attached response letter.

[Comment 2-2]

Some technical remarks: it would be great if more documentation, explanation of the individual steps, and installation instructions could be added to the Jupter Notebooks. The figures appear in very low quality in the PDF, but this might be a production issue unrelated to the submitted material.

[Response 2-2]

We have updated the Jupyter Notebooks by adding additional documentation and comments to clarify the individual steps. We have also renamed some variables and functions to make the code more intuitive. We hope these changes improve the clarity and usability of the notebooks. The updated Jupyter Notebook is publicly available on Dryad (https://doi.org/10.5061/dryad.2280gb62f).

Regarding the figure quality issue, we would like to clarify that the version we submitted to PLOS ONE included the text and images uploaded separately. The text was provided in a DOCX format, and the images were submitted as high-resolution TIFF vector files. We did not upload any PDF versions of the images. Therefore, we guess the low image quality the reviewer mentioned here may be a production issue that occurred during the review process.

---

## [Decision Letter · Decision Letter 1]

17 Sep 2024

MCount: An automated colony counting tool for high-throughput microbiology

PONE-D-24-05814R1

Dear Dr. Buie,

We’re pleased to inform you that your manuscript has been judged scientifically suitable for publication and will be formally accepted for publication once it meets all outstanding technical requirements.

Kind regards,

Florian Rehfeldt

Academic Editor

PLOS ONE

Additional Editor Comments (optional):

Reviewers' comments:

Reviewer's Responses to Questions

**Comments to the Author**

1. If the authors have adequately addressed your comments raised in a previous round of review and you feel that this manuscript is now acceptable for publication, you may indicate that here to bypass the “Comments to the Author” section, enter your conflict of interest statement in the “Confidential to Editor” section, and submit your "Accept" recommendation.

Reviewer #1: All comments have been addressed

Reviewer #2: All comments have been addressed

2. Is the manuscript technically sound, and do the data support the conclusions?

Reviewer #1: (No Response)

Reviewer #2: Yes

3. Has the statistical analysis been performed appropriately and rigorously? 

Reviewer #1: (No Response)

Reviewer #2: Yes

4. Have the authors made all data underlying the findings in their manuscript fully available?

Reviewer #1: (No Response)

Reviewer #2: Yes

5. Is the manuscript presented in an intelligible fashion and written in standard English?

Reviewer #1: (No Response)

Reviewer #2: Yes

6. Review Comments to the Author

Reviewer #1: I thank the authors for their response. They have succesfully addressed all my concerns, and I support publication.

Reviewer #2: The authors now included a more detailed comparison with existing approaches and a discussion of recent literature in the field. The dataset on which the performance is demonstrated still is rather limited and not representative of a wide range of possible user scenarios, but since the tool and source code is available, readers can try and include the algorithm in their own workflows.

7. PLOS authors have the option to publish the peer review history of their article (what does this mean? ). If published, this will include your full peer review and any attached files.

**Do you want your identity to be public for this peer review?** For information about this choice, including consent withdrawal, please see our Privacy Policy .

Reviewer #1: No

Reviewer #2: No

---

## [Editor Report · Acceptance letter]

PONE-D-24-05814R1

PLOS ONE

Dear Dr. Buie,

I'm pleased to inform you that your manuscript has been deemed suitable for publication in PLOS ONE. Congratulations! Your manuscript is now being handed over to our production team.

Kind regards,

on behalf of

Dr. Florian Rehfeldt

Academic Editor

PLOS ONE